

# An *in silico* study of how histone tail conformation affects the binding affinity of ING family proteins

Nadir Gül[1] and Ahmet Yıldız[2]

[1] Faculty of Natural Sciences, Turkish-German University, Istanbul, Turkey
[2] Faculty of Engineering, Turkish-German University, Istanbul, Turkey

## ABSTRACT

**Background**. Due to its intrinsically disordered nature, the histone tail is conformationally heterogenic. Therefore, it provides specific binding sites for different binding proteins or factors through reversible post-translational modifications (PTMs). For instance, experimental studies stated that the ING family binds with the histone tail that has methylation on the lysine in position 4. However, numerous complexes featuring a methylated fourth lysine residue of the histone tail can be found in the UniProt database. So the question arose if other factors like the conformation of the histone tail affect the binding affinity.

**Methods**. The crystal structure of the PHD finger domain from the proteins ING1, ING2, ING4, and ING5 are docked to four histone H3 tails with two different conformations using Haddock 2.4 and ClusPro. The best four models for each combination are selected and a two-sample t-test is performed to compare the binding affinities of helical conformations vs. linear conformations using Prodigy. The protein-protein interactions are examined using LigPlot.

**Results**. The linear histone conformations in predicted INGs-histone H3 complexes exhibit statistically significant higher binding affinity than their helical counterparts (confidence level of 99%). The outputs of predicted models generated by the molecular docking programs Haddock 2.4 and ClusPro are comparable, and the obtained protein-protein interaction patterns are consistent with experimentally confirmed binding patterns.

**Conclusion**. The results show that the conformation of the histone tail is significantly affecting the binding affinity of the docking protein. Herewith, this *in silico* study demonstrated in detail the binding preference of the ING protein family to histone H3 tail. Further research on the effect of certain PTMs on the final tail conformation and the interaction between those factors seem to be promising for a better understanding of epigenetics.

# INTRODUCTION

Chromatin dynamics is a rich modulation scene that is influenced by nucleosome motions as well as reversible post-translational modifications (PTM) of the histone tails, and it

Corresponding author
Nadir Gül, nadir.gul@tau.edu.tr

governs cellular viability and nuclear operations by affecting the accessibility of DNA on histone proteins (*Armeev et al., 2021*; *Huertas, Schöler & Cojocaru, 2021*).

The tail of the histone, which is an intrinsically disordered protein (IDP) (*Uversky, Gillespie & Fink, 2000*; *Van der Lee et al., 2014*), that extends from the chromatin structure, can undergo conformational changes (*Bortoluzzi et al., 2017*; *Fuchs et al., 2011*; *Musselman & Kutateladze, 2021*), thus forming a molecular recognition site for binding proteins like histone readers as well as regulating the mechanisms that specify which reader or binding factor will be equipped (*Musselman & Kutateladze, 2021*; *Peng et al., 2021*).

IDPs like Histone H3 supposedly adopt a clearly defined conformation when they interact with a target molecule (*Hansen et al., 2006*). According to *Peng et al. (2021)*, binding factors and histone tails interact through competitive attachment or tail displacement mechanisms.

However, lysine (K) residues within histone tails are reversibly modified through the addition of methyl or acetyl groups (*Taverna et al., 2006*; *Li & Li, 2012*). These specific modification patterns commonly converge to form or displace specific binding sites for other proteins. Supporting this view, experimental studies show that hyperacetylation of histone tail increases their average $\alpha$-helical content (*Wang et al., 2000*) Moreover; in X-ray and NMR studies methylated histone tail adopts an extended chain structure, actually serving to fill in a $\beta$-sheet (*Nielsen et al., 2002*; *Fischle et al., 2003*). This chain structure of the histone tail provides an optimal binding site.

The histone readers are a group of diverse proteins that recognize and bind specifically to the N-tail of histones leading to chromatin remodeling or involving in gene expression or joining as chromatin architectural proteins. One of these histone readers, the Inhibitor of Growth (ING) protein family is substantially conserved across all eukaryotes implying that they may contribute to critical biological processes and may also have complementary functions due to their similarities (*Cui et al., 2015*). Genuinely, ING proteins, like other tumor suppressor factors, are involved in essential processes such as apoptosis, DNA repair, and cellular senescence, thus they've aroused interest (*Larrieu et al., 2009*; *Cheung Jr et al., 2001*; *Wang, Chin & Li, 2006*). Further, emerging studies demonstrate that ING proteins, as well as the complexes they associate with other proteins, play a pivotal role in transcription regulation and epigenetic regulation (*Ormaza et al., 2019*; *Unoki et al., 2009*).

Many regions in the protein structure are thought to be essential for the function of ING proteins. The plant homeodomain (PHD)-like zinc finger domain is the most conserved region located at the C-terminus of ING proteins and is involved in chromatin remodeling through interaction with specific nuclear protein partners (*Champagne & Kutateladze, 2009*; *He et al., 2005*; *Jacquet & Binda, 2021*; *Bienz, 2006*; *Ragvin et al., 2004*). On the other hand, the N-terminus differs amongst ING members and determines their specific functions, along with antagonistic regulatory characteristics (*Kataoka et al., 2003*). Therefore, members of the ING gene family have been demonstrated to have diverse epigenetic functions (*Tallen & Riabowol, 2014*; *Doyon et al., 2006*). They function as histone readers, core components of histone deacetylases (HDACs) 1 and 2, and chromatin-modifying complexes, including histone acetyltransferase (HAT), monocytic leukemia zinc finger protein, and the related factor (MOZ/MORF). Further, INGs influence

cancer hallmarks through modulating gene methylation patterns, primarily as tumor suppressors (*Tallen & Riabowol, 2014*).

The PHD finger is a region that varies from 50 to 80 amino acids and contains a zinc-binding motif (*Aasland, Gibson & Stewart, 1995*). The most conserved property is the binding of H3's first six N-terminal residues (ARTKQT) to the PHD finger's two-strand $\beta$-sheet ($\beta1$ and $\beta2$) *via* the formation of an antiparallel-strand. A two-strand anti-parallel ß-sheet and a C-terminal -helix (not present in all PHDs) are stabilized by two zinc atoms bound by the Cys4-His-Cys3 motif in a cross-brace architecture in the conserved PHD fold (*Li et al., 2006*; *Kwan et al., 2003*). The C terminal PHD finger domain of INGs binds strongly and specifically to the N-tail of histone H3 with an increased affinity for the methylation status of the 4th positioned lysine amino acid (H3K4) (*Champagne & Kutateladze, 2009*; *Soliman & Riabowol, 2007*; *Champagne et al., 2008*; *Ali et al., 2012*; *Palacios et al., 2008*; *Shi et al., 2006*; *Peña et al., 2006*).

Nevertheless, growing evidence suggests that histone tails modulate the accessibility of binding DNA, as well as the accessibility of binding components in solvent (*Musselman & Kutateladze, 2021*; *Morrison et al., 2018*). In this context, interactions between the ING PHD finger domain and the histone H3 tail are influenced by individual binding affinity of proteins and histone tail conformational dynamics. It has been noted that by changing the electrostatics of the tail with modifications or mutations occurring in the histone tail, the accessibility of PHD to the histone binding site increases and thus modulates the binding (*Musselman & Kutateladze, 2021*; *Morrison et al., 2018*).

In the RCSB protein database (https://www.rcsb.org/) (*Palacios et al., 2008*), there are 17,191 complexes only with H3k4me3. H3k4me3-PHD finger complexes contribute 2,168 of these complexes, whereas H3k4me3-ING proteins account for 162 structures. The ability of the histone H3 N-terminal tail to form a wide variety of complexes with just one modification has drawn the main attention of researchers to this me3-modification as a very important indicator for building complexes. However, to the knowledge of the authors, there are not many studies investigating the conformational structure within these complexes and no studies at all about the affinity of ING proteins to bind to specific conformations.

Following this initial information, to better understand if and how histone H3 conformations affect ING binding affinity, an *in-silico* investigation including computational molecular docking was performed in this study. Four different computationally derived histone conformation models were constructed to investigate the binding affinity between histone H3 and INGs. Despite minor differences, two of these four conformations are helical, and the other two are linear conformations containing the ARTKQTARKST (H3-11) sequence.

Along with increasing computational capabilities, molecular docking studies on protein-protein interaction mainly constitute predictive models with steric and physicochemical properties at the protein interface. These calculations use biochemical and biophysical interactions resulting from NMR titration experiments or mutagenesis data (*Vakser, 2014*; *Dominguez, Boelens & Bonvin, 2003*).

Various *in-silico* approaches, like molecular simulations or molecular docking along with statistical calculations, have been used to broaden the boundaries of experimental capabilities and make it easier to understand complex structures due to the complexity of biological materials (*Papamokos, 2019*).

Due to the limitations of molecular simulation techniques (*Ikebe, Sakuraba & Kono, 2016*), we preferred molecular docking methods. Molecular docking systems provided us the ability to examine the interactions between various conformations of the histone tail and the ING protein, making it possible to perform a statistical analysis of the results. The present paper is to the best of our knowledge one of the first *in-silico* attempts in this field and our findings highlight the importance of the conformation of the histone tail. These results should lead to further research with the goal of better understanding the nature of those different conformations and their relation with the well-studied PTMs.

## MATERIALS & METHODS

### Structural design

The crystal structures of the ING proteins utilized in this study, which are summarized in Table 1, were gathered from the protein database (http://www.rcsb.org/) (*Berman et al., 2000*). Additionally, four (two linear, two helical) different three-dimensional structures of the histone H3 N-Tail (11 amino acid) conformations were modeled using UCSF Chimera (*Pettersen et al., 2004*). Figure 1 shows the 3-dimensional structure of these four conformations.

As there was no crystal structure for the ING2 protein derived from humans in the database, it was decided to use the structure from a mouse, because there are no sequential differences in the crystal structures of the PHD Finger domain (213–262 positions) between ING2 human (UniProtKB-Q9H160) and ING2 mouse (UniProtKB-Q9ESK4) proteins. Since there are no experimentally verified crystal structures of the ING3 protein, it could not be included in this study.

### Multiple alignment & structure comparison

Sequential information was retrieved from UniprotKB (http://www.uniprot.org/) and multiple sequence alignment of the proteins was carried out in Clustal Omega (https://www.ebi.ac.uk/Tools/msa/clustalo/). JalviewV2 was used for the visualization of the alignment results (*Waterhouse et al., 2009*; *Tian et al., 2018*).

Comparison analyses of the three-dimensional structures and their molecular visualizations were performed using UCSF Chimera (*Pettersen et al., 2004*).

### Molecular docking analysis

The geometric and topological features of protein architectures are critical for proteins to execute their interactions. In this context, as a preparation step to improve a more consistent docking, the active and passive binding regions of ING proteins were detected with the Computed Atlas of Surface Topography of Proteins (CASTP) online service (*Tian et al., 2018*).

**Table 1  Data snapshot of ING proteins utilized in the study.** The crystal structures of the ING proteins were gathered from the protein database (http://www.rcsb.org/).

| Protein/Domain name | PDB ID | Sequence length | Resolution | Model | Ref. |
|---|---|---|---|---|---|
| ING1 PHD finger (human) | 2QIC | 62 | 2.10 Å | X-RAY Diff. | *Papamokos (2019)* |
| ING2 PHD finger (mouse) | 2G6Q | 62 | 2.00 Å | X-RAY Diff | *Ikebe, Sakuraba & Kono (2016)* |
| ING4 PHD finger (human) | 2VNF | 60 | 1.76 Å | X-RAY Diff. | *Pettersen et al. (2004)* |
| ING5 PHD finger (human) | 3C6W | 59 | 1.75 Å | X-RAY Diff. | *Waterhouse et al. (2009)* |

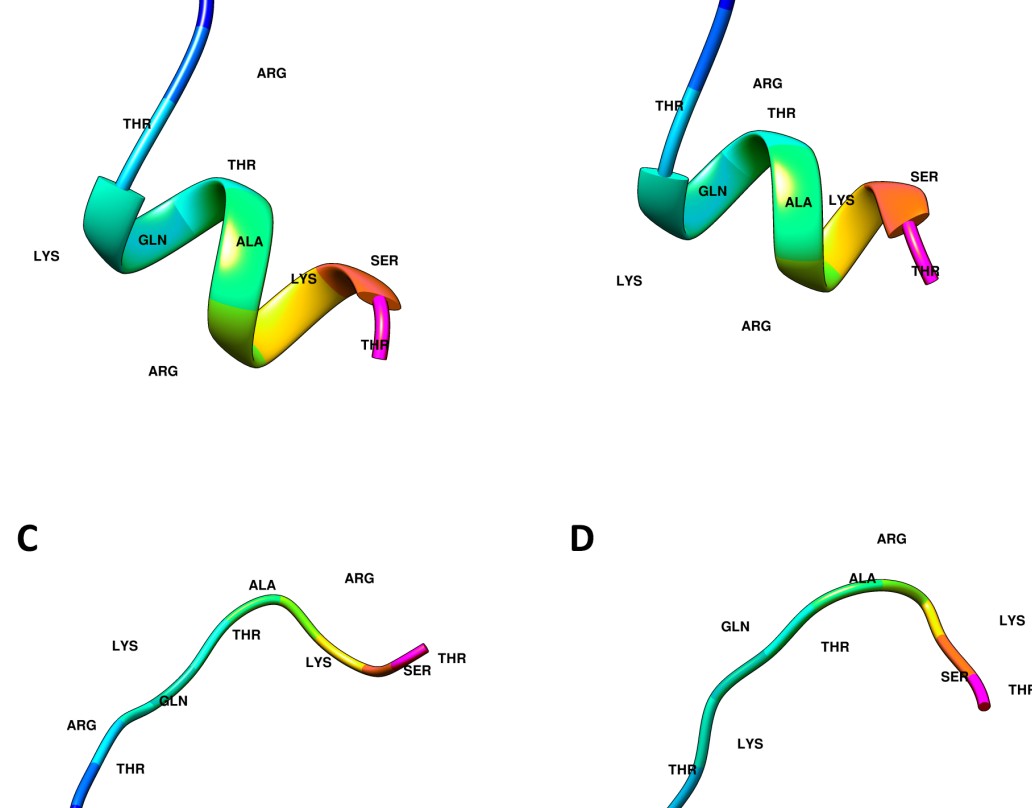

**Figure 1  Computational generated histone H3 N-tail conformations.** Histone tail secondary structure models in the helical (A, B) and linear (C, D) conformations are illustrated with colored residues along the rainbow color scale from N terminal (blue) to C terminal (magenta).

*In-silico* prediction of INGs to histone H3-N tail interactions was achieved through two different online services, ClusPro and Haddock 2.4. Thus, by using two systems that implement different docking algorithms the results can be cross-checked.

ClusPro is based on the rigid body docking algorithm PIPER (*Kozakov et al., 2006*), created using the Fast Fourier Transform (FFT) correlation approach, which generates and evaluates countless models of INGs-H3 N-tail complexes. The complex structures were then sorted using nine Å C-alpha radius pairwise root-mean-square deviations (RMSD) as the distance metric.

The server returns 10 different complexes as result, which were ranked according to the cluster size and lowest energy (*Desta et al., 2020*; *Vajda et al., 2017*).

Haddock 2.4 is an information-driven flexible docking approach for modeling. Haddock 2.4 implements the topology of the molecules to be docked automatically. For the production of ambiguous interatomic restraints (AIRs) in Haddock 2.4, the CASTP service was utilized to define the active and passive residues. This information is used by Haddock 2.4 to compile topology files. The docking methodology is then broken down into three stages: rigid body energy minimization, semi-flexible refinement in torsion angle space, and explicit solvent refinement. Constructions are rated and ranked after each of these phases, and the finest structures are retained for the following round. The Haddock 2.4 score is a weighted combination of van der Waals, electrostatic, desolvation, and restraint violation energies and as well as buried surface area (*Honorato et al., 2021*; *Van Zundert et al., 2016*; *De Vries, Van Dijk & Bonvin, 2010*).

The binding affinity for the histone H3-ING complexes was then calculated through the PROtein binDIng enerGY prediction (PRODIGY) web-server (*Vangone & Bonvin, 2017*; *Xue et al., 2016*). PRODIGY is an online application that predicts the binding affinity and dissociation constant of biological complexes using an atomic contacts-based prediction method.

For each of the 16 ING-histone combinations, the best four models both from ClusPro and Haddock 2.4 were chosen for further analysis. This resulted in 32 values for binding affinities and dissociation constants per histone protein, or 64 values per conformation type (helical or linear). With these values, a two-sample $t$-test was a feasible method to check the statistical significance of the differences.

Finally, LIGPLOT plus was used to produce schematic diagrams to easily examine protein-protein interactions in docking complexes. LIGPLOT plus is a software for drawing schematic linear representations of protein chains in terms of their structural domains (*Wallace, Laskowski & Thornton, 1995*).

## RESULTS

### Structural differences of the PHD finger

The PHD finger domains are known to be protected regions. Literature shows that proteins like MORF or DPF3 are building complexes with histone H3 proteins' helical tails (*Klein et al., 2017*; *Local et al., 2018*). Therefore, the PHD finger domains of these two proteins and the four ING proteins investigated in this study are compared with a multiple sequence alignment performed in Clustal Omega. The similarities and differences visualized with JalviewV2 (*Taverna et al., 2006*; *Li & Li, 2012*) can be seen in Fig. 2.

The small differences in the sequence of the PHD domain of the four investigated ING proteins results in structural differences as well, which are larger. Figure 3 shows
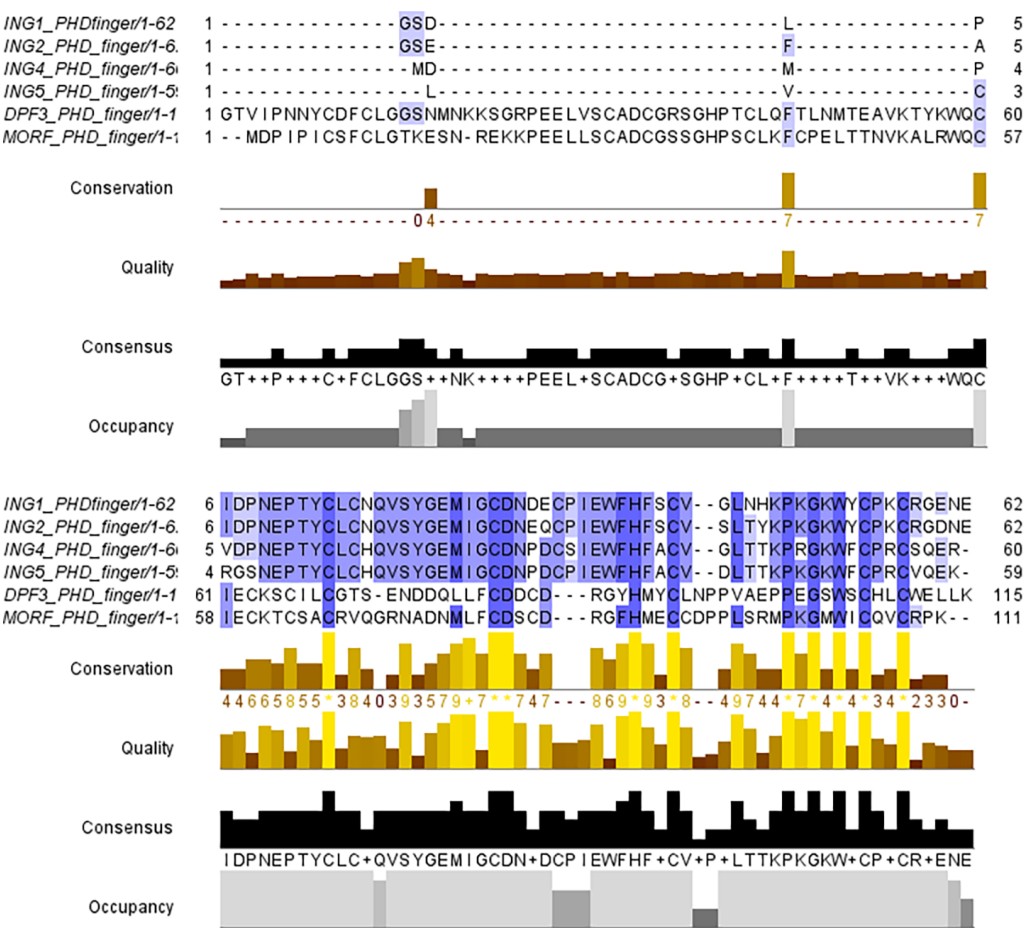

**Figure 2** **Multiple sequence alignment of PHD finger domains.** Multiple sequence alignment of PHD fingers of ING1 (PDB: 2QIC), ING2 (PDB: 2G6Q), ING4 (PDB: 2VNF), ING5 (PDB: 3C6W) PHD finger of MORF (PDB: 5U2J) and PHD finger of DPF3 (PDB: 5SZB). Similar residues are shown as purple color gradients when the percent identity of the INGs PHD region is compared to the MOZ and DPF PHD sections, which are known to bind to the histone tail in a helical conformation.

the superposed structures of the PHD domains. Table 2 gives the similarity percentage of the combinations, with a very high similarity between ING1-ING2 as well as between ING4-ING5.

Additionally, the MOZ and DPF3 PHD fingers, which have been experimentally verified to bind with histone in a helical conformation, differ significantly from the ING PHD fingers in terms of both sequence patterns and sequence lengths. These differences suggest that PHD fingers could be able to account for the different affinities of histone proteins in respective conformations.

## Linear conformations are preferred while docking

To answer the research question, of whether the structural conformation of the histone H3 tail affects the binding affinity of ING proteins, *in-silico* protein-protein interactions were calculated. For each combination, the four best models with the lowest energy and lowest

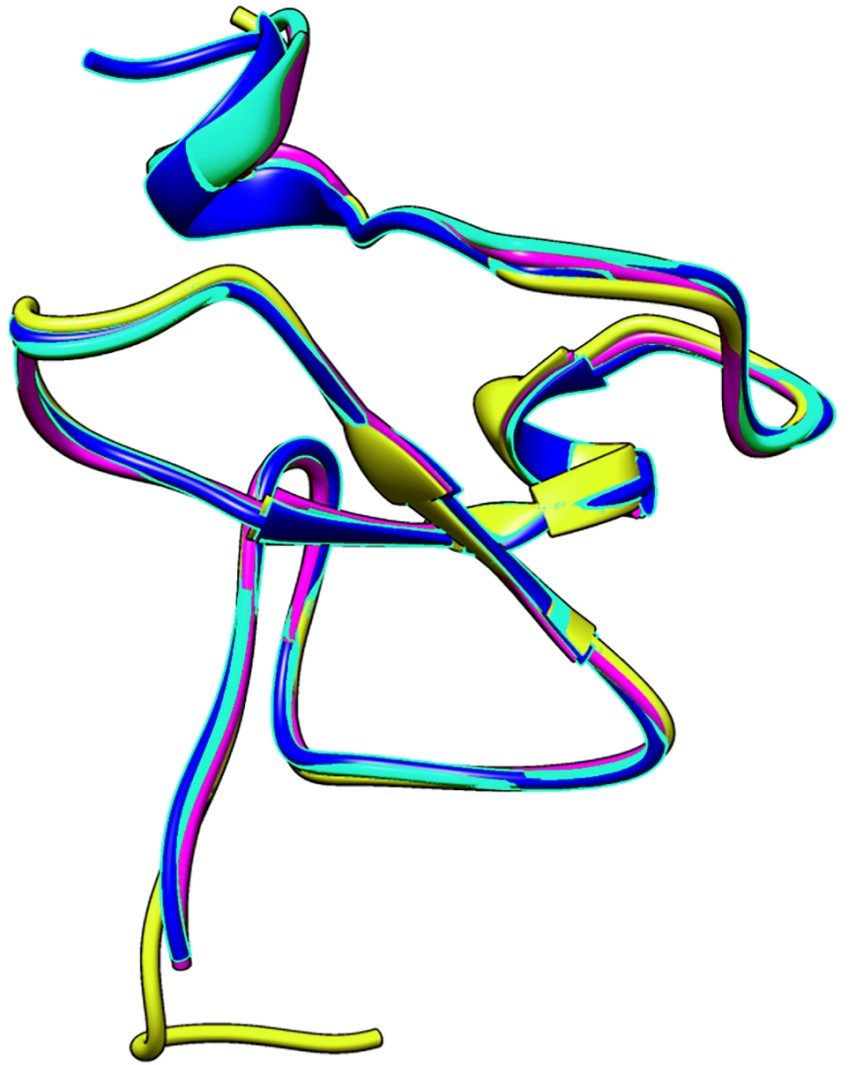

**Figure 3** The superposed PHD finger domains of ING1 (Magenta), ING2 (Light blue) ING4 (yellow), ING5 (blue).

**Table 2** Structural comparison of PHD finger domains of ING proteins. Comparison of structural similarity.

|  | ING2 | ING4 | ING5 |
| --- | --- | --- | --- |
| ING1 | 88.24% | 74.51% | 76.47% |
| ING2 |  | 73.08% | 78.43% |
| ING4 |  |  | 90.2% |

Z-Score were selected. For all models, the binding affinity and the dissociation constant were calculated with Prodigy. The results can be seen in Fig. 4.

One can see that the linear conformations always have lower energies than the helical conformations, while the smallest difference is for the ING2 protein. As a point of reference,

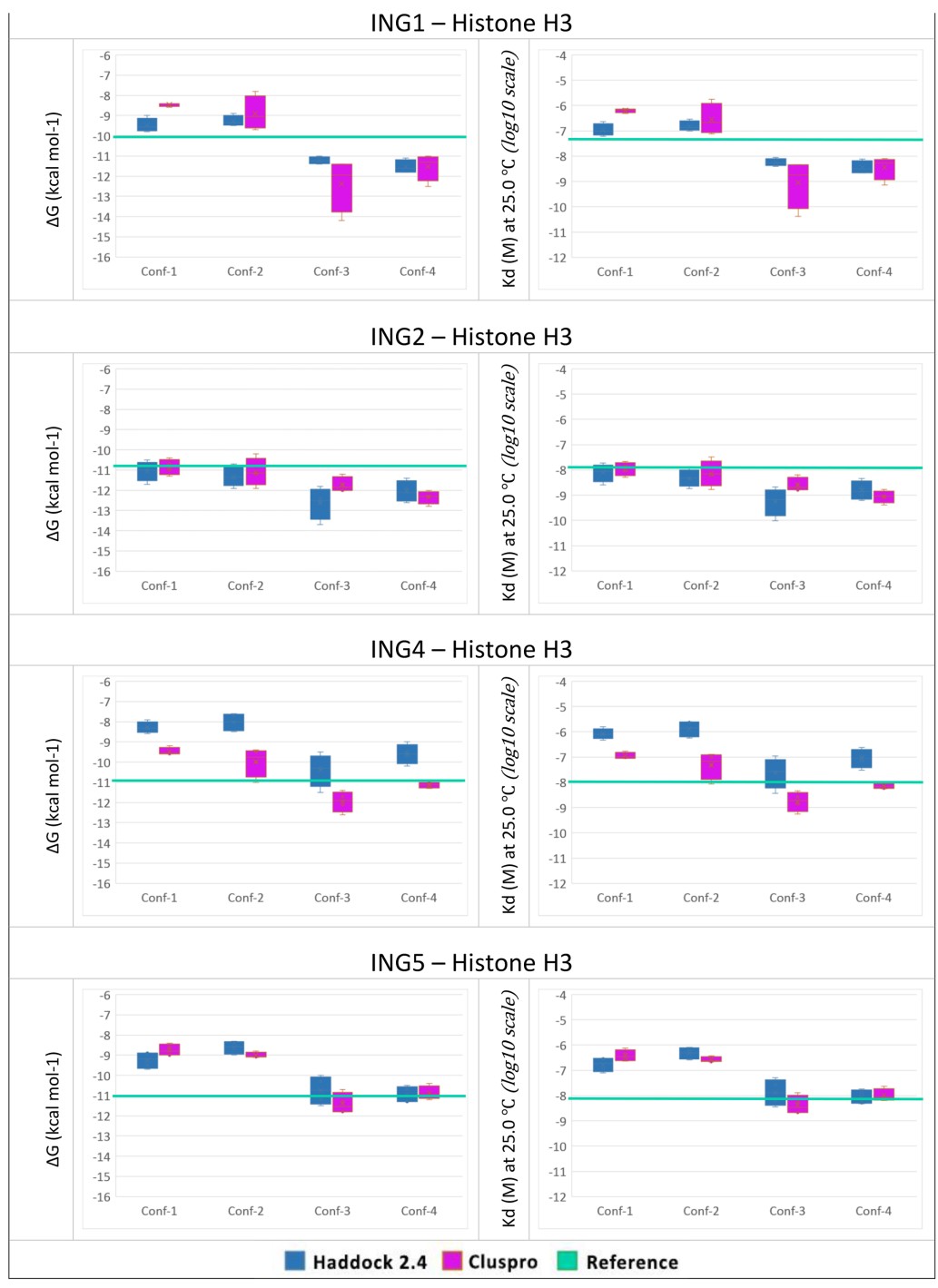

**Figure 4  PRODGY binding affinity analyses.** Binding affinity analysis of ING proteins on different histone H3 tail conformations. Helical conformations (conformation 1 and 2) Linear conformations (conformation 3 and 4) X-ray experimental structures have been utilized as reference value (green line) docking results depicted from Haddock in blue and ClusPro in magenta. All docking outcomes were statistically calculated with reference values. Significantly lower binding energies are seen in INGs-Histone (linear) complexes compared to INGs-Histone (helical) complexes.

the same values were calculated with Prodigy for the original crystallographic PDB data. These reference lines are shown in green in Fig. 4. For ING1 the reference value represents a clean cut between linear and helical conformations, whereas for ING4 and ING5 it is more of a median of the linear conformations. Only ING2 has a very high reference value directly overlapping with helical conformations.

The two-valued T-Test was performed for each ING protein by dividing the calculated binding affinity for helical and linear conformations. Using a 99% confidence interval the $p$-value was always less than 0.0001.

All calculations are confirming the difference between the two histone H3 tail models' affinity to bind with ING proteins with a clear preference for linear conformations over helical ones.

### Similar binding patterns in different conformations

For a better understanding of the results, the actual connections between the amino acids of the four ING proteins and the histone H3 tail were investigated using Ligplot. The result is illustrated in Fig. 5. These calculated connections are in accordance with the experimentally verified connections in the crystallographic data.

To examine the binding analysis between INGs and histone H3 predicted models in more detail, the binding analysis with Ligplot was performed with two models of ING1 with the best binding affinity. For the linear histone H3 tail, the G value was −11.4 kcal mol-1 and the Kd (M) value was 4.00E−09 at 25.0 °C. For the helical histone H3 tail, the values were −9.8 kcal mol-1 and 6.10E−08 at 25.0 °C respectively. The interactions are schematically illustrated for the linear conformation in Fig. 6 and the helical conformation in Fig. 7.

It is known that ING PHD recognizes the histone tail through K4 and forms a complex with the participation of R2. Similar to the results of experimental studies, our calculated models show that in both linear and helical conformations, ING PHD-histone H3 complex is also mainly based on these two amino acids.

Figure 6 shows that the linear histone tail provides an optimal surface for the binding of the rigid ING PHD, and seven amino acids (A1, R2, T3, K4, T6, R8, K9) in the 11 amino acid histone tail provide hydrogen bonding and hydrophobic interactions. On the other hand, these interactions are limited to four amino acids in the histone tail in helical model. This explains the importance of the histone tail conformation for selecting binding proteins.

## DISCUSSION

One of the most important structures that can be held responsible for gene transcription in chromatin dynamics is the histone tail, which protrudes from the nucleosome and can change conformation through its reversible modification by many chemical groups (*Armeev et al., 2021*). These chemical groups act as epigenetic patterns, enabling the binding partner to bind to the histone tail specifically.

Previous experimental studies have suggested that the PHD finger of ING proteins can recognize the histone H3 tail with varying affinities depending on the methylation status

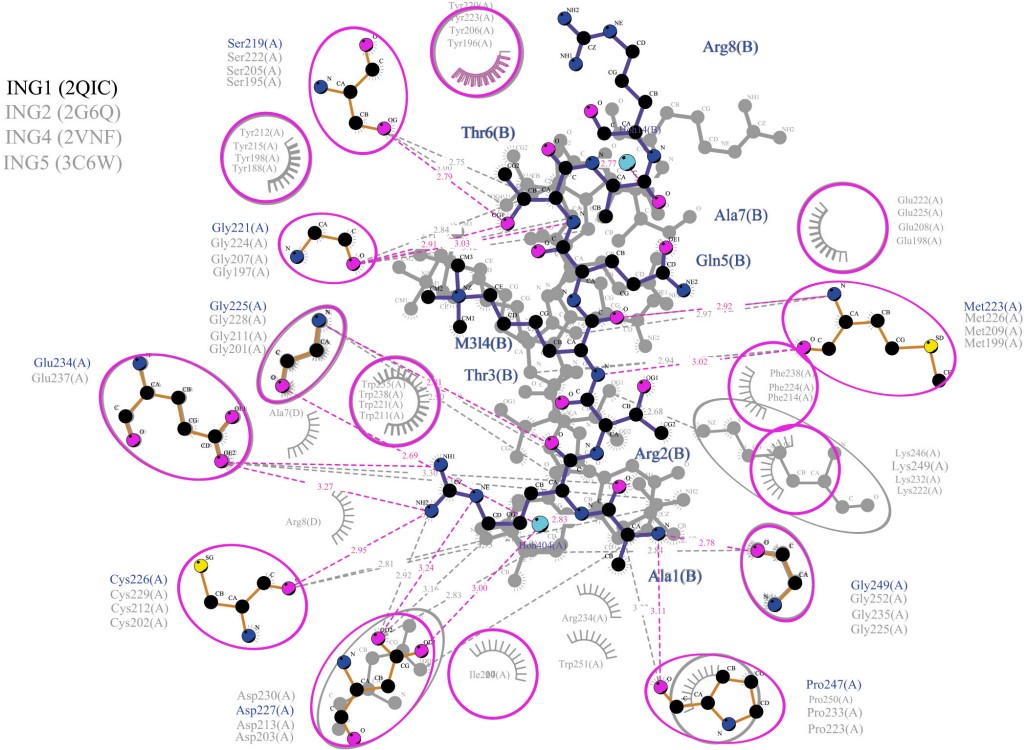

**Figure 5  Merged illustration of INGs–Histone H3 complexes.** Merged illustration of INGs H3K4me3 complex from x-ray crystal structures. Bindings and interactions between the histone tail's at the centre and the ING PHD regions surrounding the histone tails (same residues in different INGs are grouped in magenta circles).

of the K4 (*Champagne & Kutateladze, 2009*; *Soliman & Riabowol, 2007*; *Peña et al., 2006*) as well as unmodified or various modifications like acetylation and phosphorylation (*Li & Li, 2012*; *Musselman & Kutateladze, 2011*; *Papamokos et al., 2021*). The binding site of the PHD finger grips the K4 of the histone H3 tail while the R2 is coordinated in a neighboring pocket. A small residue is needed at position 3 of the histone tail to form the narrow channel that connects these two sites  (*Kwan et al., 2003*; *Kim et al., 2016*). Further, the molecular simulation revealed that the formation of an ING—histone H3 complex is driven by a combination of hydrogen bonding as well as hydrophobic contacts and surface interactions (*Kim et al., 2016*). Nevertheless, these studies did not reflect on the selectivity of the conformational change in the histone tail whereas they emphasized the modifications in the histone tail.

The histone proteins display a significant conformational heterogeneity and do not have an equilibrium geometry. However, the structure of the conformation is never random and the IDPs have some preferential conformations (*Dunker et al., 2013*).

NMR and all-atom MD simulation studies indicate that the unmodified N-tail of histone is intrinsically disordered. Studies suggest that the helical conformation of the histone tail can be regulated by PTMs, mostly by the acetylation of the lysines, while
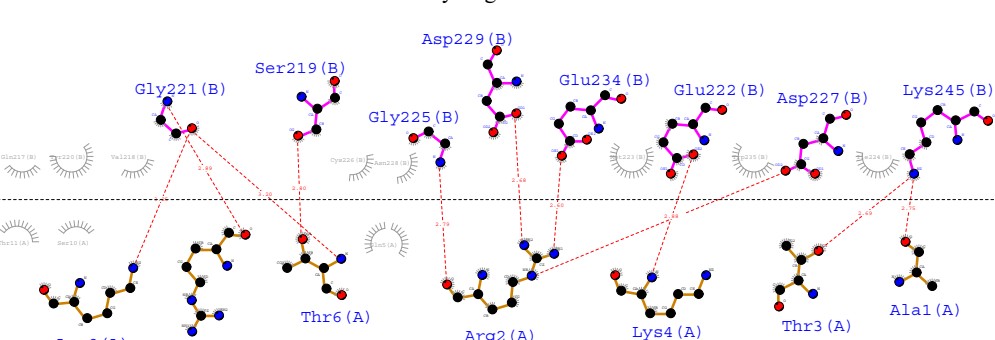

ING1-Histone H3 (Linear)
Hydrogene Bonds

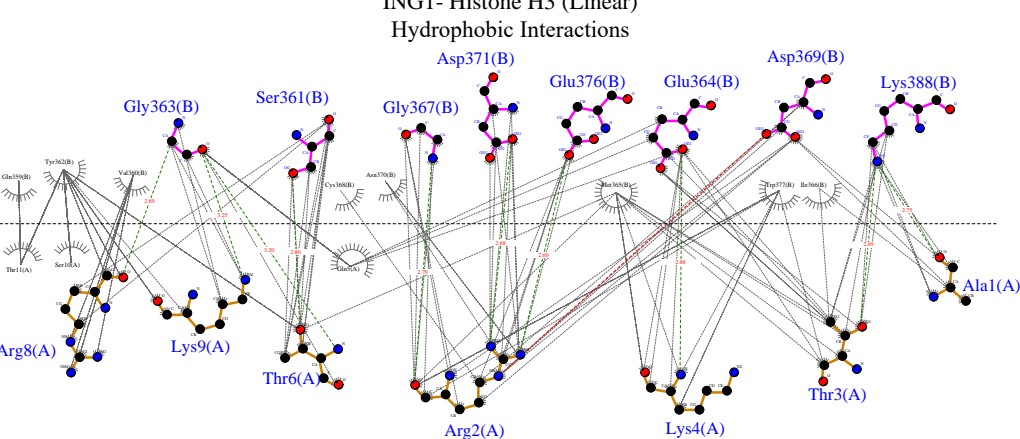

ING1- Histone H3 (Linear)
Hydrophobic Interactions

**Figure 6** **Ligplot diagram of ING1-Histone H3 (linear) complex.** Bindings (H bond) and hydrophopıc ınteractions of ING1-Hıstone H3 (linear) complex. A favourable binding surface for ING PHD is provided by the linear histone tail conformation for both their H bonding (upper image) and their hydrophobic interaction (lower image).

circular dichroism studies showed that the histone tail adopts a helical conformation 50% of the time (*Ghoneim, Fuchs & Musselman, 2021*).

*Bortoluzzi et al. (2017)* identified three possible conformations in which the histone tail builds a complex with a PHD finger; helical, bent and fully extended. Moreover, they explained that BAZ2a H3 assumes a helical fold when in complex with PHD fingers that harbor a short helical turn or loop just before the first β-strand (*Van der Lee et al., 2014*). Further studies identified the PHD fingers MORF, MOZ, and DPF forming complexes with the α-helical conformation of Histone H3 (*Klein et al., 2017*; *Local et al., 2018*; *Dreveny et al., 2014*).

In the light of these experimental data, we tested the consistency of our *in-silico* methodology by performing the same calculations with MOZ PHD—histone H3 complexes (PDB: 4LK9) (64). The *in-silico* calculations showed a statistically significant higher affinity

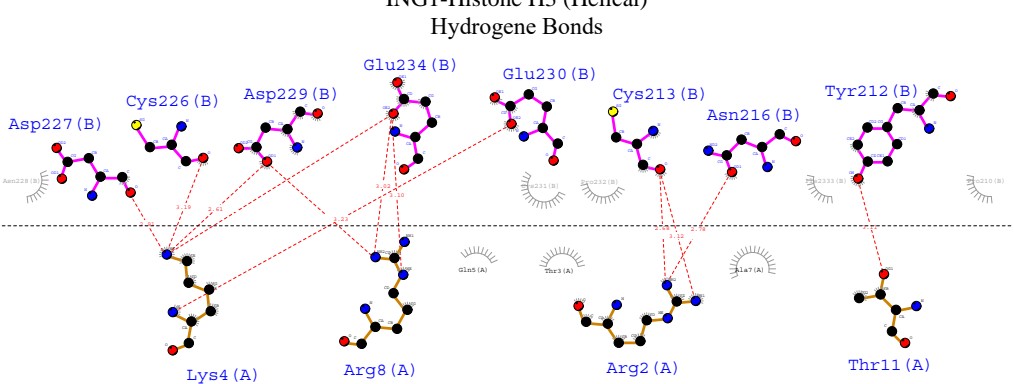

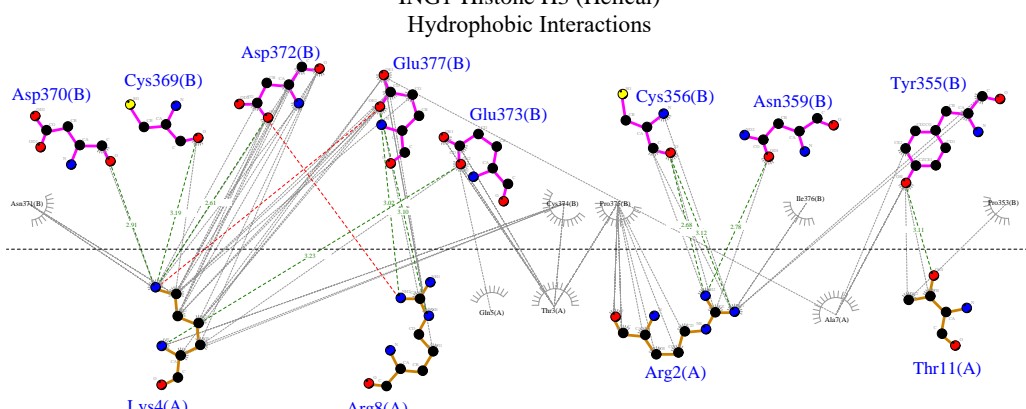

**Figure 7  Ligplot diagram of ING1-Histone H3 (helical) complex.** Bindings (H bond) and hydrophopıc interactions of ING1-Histone H3 (helical) complex. As a consequence of the helical geometry, almost half of the histone tail remains away from the ING PHD binding site, theoretically forming a weak complex. K4 and R2 play a major role in the complexing in helical conformation as well as in linear conformation.

for binding with helical histone tail models. These outcomes which are in line with all cited experimental results indicate the reliability of our constructed methodology.

Importantly, studies imply that the PHD finger of INGs does not undergo any conformational changes during binding (*Taverna et al., 2006*; *Li et al., 2006*; *Kwan et al., 2003*). This was taken into account both with Haddock 2.4 and with ClusPro while performing the dockings. Coherent with experimental data, our predictive models also show binding to the K4 and R2 residues, regardless of the conformation of the histone tail.

The binding analysis with ligplot displays that the S219, G221, M223 G225, C226, D227, E234, G249, and P247 amino acids of the ING1 PHD finger build hydrogen bonds with histone H3$_{linear}$ A1, R2, T3, K4, T6, R8 and K9. The best scored model of the ING1-H3$_{linear}$ complex showed very similar bonds in number and shape as the experimentally verified X-ray crystallographies.

However, the calculated model of the ING PHD and histone H3$_{helical}$ complex relied on hydrogen bonds and hydrophobic interactions of the R2, K4, R8, and T11 amino acids of the histone tail. Due to the helical form of the histone tail, the residues which are located on the outer side of the helix were not reachable by the PHD finger of the ING proteins.

When we look at the binding affinity and dissociation constants in both complexes, one can see that the ING-H3$_{linear}$ complexes always have lower energies than the ING-H3$_{Helical}$ complexes. As a point of reference, the same values were calculated with Prodigy for the original crystallographic PDB data. The reference value represents a clean cut between linear and helical conformations for ING1 and is most coherent with the calculated values for linear conformations.

These results suggest that the rigid structure of the ING PHD finger does not prefer to bind to the histone H3$_{Helical}$ conformation. This led to the conclusion that besides PTM like H3K4me3, the conformation of the histone H3 tail has also an important influence on the selectivity of ING proteins.

## CONCLUSIONS

The main question motivating this research was the effect of the histone tail conformation on the binding affinity of proteins. The ING family was chosen as an important protein. Existing experimentally verified data and most of the literature concentrated on different PTMs while disregarding the conformation of the histone tail.

The performed *in-silico* calculations showed, that there is indeed a statistically significant difference between the binding affinity depending on the conformational shape of the histone tail.

Further research is needed to better understand the mechanisms and also the possible relation between PTMs and the final conformation of the histone tail.

## ACKNOWLEDGEMENTS

We want to thank Dr. Seher Karslıfrom Marmara University for her invaluable guidance and initial help leading to this final paper and Assoc.Prof.Dr.rer.nat. Tuba Çonka Yıldız from Turkish-German University for proofreading. We also want to thank our editor and our reviewers. Their invaluable feedback made it possible to make this research paper more profound and also set the goals for future research activities. This research was done by the author Nadir Gül while he was a guest lecturer at Turkish German University.

### Funding

The authors received no funding for this work.

### Competing Interests

The authors declare there are no competing interests.

## Author Contributions

- Nadir Gül conceived and designed the experiments, performed the experiments, analyzed the data, prepared figures and/or tables, authored or reviewed drafts of the article, and approved the final draft.
- Ahmet Yıldız analyzed the data, prepared figures, authored or reviewed drafts of the article and approved the final draft.

## Data Availability

The raw data is available in the Supplemental Files.

## Supplemental Information

Supplemental information for this article can be found online at http://dx.doi.org/10.7717/peerj.14029#supplemental-information.

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
