# Peer review of "An in silico study of how histone tail conformation affects the binding affinity of ING family proteins"

_PeerJ, doi:10.7717/peerj.14029_

## Round 0.1 · original submission · Major Revisions

Please address the concerns of all reviewers and revise the manuscript accordingly.

Reviewer 1 ·

Basic reporting

Figures need more details. For example, what are the meanings and indications of all the colored squares in figure 4? What are the meanings of the red circles in figure 5? Please elaborate all figure legends.

Figure 1 and table 1 were not referenced in main text.

Experimental design

no comment

Validity of the findings

no comment

Additional comments

no comment

Reviewer 2 ·

Basic reporting

see below

Experimental design

See below

Validity of the findings

See below

Additional comments

The work of Gul and Yildiz, entitled "An in-silico study of how histone tail conformation affects the binding affinity of ING family proteins," is an interesting study of the interaction between the H3 oligopeptide and the ING family of proteins.

The paper would be more interesting if the authors could discuss, based on the literature and their results, the ability of the intrinsically disordered oligopeptide H3 (11 amino acids) to form helices. Intrinsically disordered proteins do not adopt a specific secondary conformation. Even if they are built as α-helices, they do not retain this structure when they are free. A good example is the work below:
Papamokos et al. "Progressive Phosphorylation Dictates the Self-Association of a Variably Modified Histone H3 Peptide". Frontiers in Molecular Biosciences 2021, 8, Article no 698182
where an almost identical oligopeptide is used.

The authors should compare their results with the above work, and other theoretical and experimental works, some of which are referenced in their paper, and highlight the ability of this IDP to adopt α-helical conformations when bound because the adaptability of IDPs is of critical role. The tools they have used give them many ways to highlight this ability.
Moreover, the ability of the IDPs to adopt various structures should also be connected to the complexity of the biological material: see Papamokos The nature of biological material and the irreproducibility problem in biomedical research. EMBO J. 2019 e101011 This would make the paper far more interesting.
Critical literature from pioneers of the IDPs field, such as Tompa, Pappu, and Dunker, should be studied, selected, and discussed in the text. The authors' results should be related to this literature. They also report literature with experimental results for α-helical structures of H3 oligopeptide when bound. It would be a significant finding if the authors could explain why the helical structures are formed based on their results and the experimental data.
The authors have used several well-known computational tools that make their works reliable. However, docking calculations followed by MD calculations would give a clearer picture. To this end, I suggest that MD simulations should be performed. The authors should discuss the paper:
Kim S, Natesan S, Cornilescu G, et al. Mechanism of Histone H3K4me3 Recognition by the Plant Homeodomain of Inhibitor of Growth 3. J Biol Chem. 2016;291(35):18326-18341. doi:10.1074/jbc.M115.690651
which includes a theoretical study of the system they are studying!
In general, the work can be published after major revision.

The language, the grammar, and syntax of the text should be substantially improved: The paper is not too long, and sloppy mistakes are not excused: e.g., in line 81, "In this context, Interactions… is…" or line 85: "…histone tails binds…" in-silico" should be formatted in italics.


The text in the introduction and discussion is repetitive.

Reviewer 3 ·

Basic reporting

In the manuscript by Gül et al., the authors used computational methods to study how the structural conformations of histone tails affect the binding of ING proteins. The alpha helical and linear conformations of the histone tails were compared. The results suggest that the ING proteins bind more tightly to the histone tails with linear conformations than helical conformations. This study is overall straightforward. However, the figure quality and the writing need to be improved.

1. The writing of this manuscript is poor. In the result section, the authors should explain why a specific modeling/calculation was performed. Then, briefly explain how it was performed. Finally, what conclusion can be made from the modeling/simulation.

2. The paragraph title of the result section should summarize findings or conclusions. In the manuscript, the authors used “Alignment of PHD”, “Docking Results”, and “Interactions” as the paragraph titles. They only explain what the authors did, but don’t explain what findings were acquired from the modeling. The authors need to modify the titles.

3. All figures should be called in the result section. Figure 1 is called in the Method section instead.

4. The first three paragraphs of the Discussion section is a repetition of the introduction and is redundant. In the first paragraph of the Discussion, the authors should briefly summarize the findings instead.

5. The figure quality is low. In Figure 1, the color coding need to be explained in the figure legend.

6. In Figure 2, remove the unnecessary red square around “C” and the black highlight of "MORF_PHD_finger".

7. In Figure 4, the font is too small for readers.

8. In Figure 5, the authors need to explain the meaning of the red circles.

9. In figure 6 and 7, the fonts are too small.

Experimental design

The manuscript fits the scope of PeerJ. The research question was well defined.

Validity of the findings

The computational data supports the conclusion. However, the conclusion needs to be experimentally tested in future study.

---

## Round 0.2 · Minor Revisions

Please address a remaining concern of reviewer #2 and amend manuscript accordingly.

Reviewer 1 ·

Basic reporting

no comment

Experimental design

no comment

Validity of the findings

no comment

Additional comments

The authors have addressed my concerns, and the manuscript is good for acceptance.

Reviewer 2 ·

Basic reporting

The manuscript can now be published:
One minor point:

The authors have added the following paragraph (lines:159-161 of the revised text):
"The complexity of biological materials makes it hard to perform reproducible experiments and favors the use of in-silico calculations and molecular simulations along with statistical calculations (42). "

Please rephrase the text and point out at least the importance of in silico calculations for the advancement of science. In the current form, the reader may think that reproducible experiments are not possible!

Experimental design

Accepted

Validity of the findings

Valid

Reviewer 3 ·

Basic reporting

The authors have sufficiently addressed all the concerns.

Experimental design

NA

Validity of the findings

NA

---

## Round 0.3 · accepted · Accept

Thank you for addressing the remaining issue of the reviewer. Your revised manuscript is acceptable now.

dd[as